# A Review of Healthy Dietary Choices for Cardiovascular Disease: From Individual Nutrients and Foods to Dietary Patterns

**DOI:** 10.3390/nu15234898

**Published:** 2023-11-23

**Authors:** Wenjing Chen, Shuqing Zhang, Xiaosong Hu, Fang Chen, Daotong Li

**Affiliations:** National Engineering Research Center for Fruit and Vegetable Processing, Key Laboratory of Fruits and Vegetables Processing, College of Food Science and Nutritional Engineering, Ministry of Agriculture, Engineering Research Centre for Fruits and Vegetables Processing, Ministry of Education, China Agricultural University, Beijing 100083, China; chenwenjing@cau.edu.cn (W.C.); zhangshuqing@cau.edu.cn (S.Z.); huxiaos@263.net (X.H.)

**Keywords:** cardiovascular disease, risk factors, macronutrients, food products, dietary patterns

## Abstract

Cardiovascular disease (CVD) remains the first cause of mortality globally. Diet plays a fundamental role in cardiovascular health and is closely linked to the development of CVD. Numerous human studies have provided evidence on the relationship between diet and CVD. By discussing the available findings on the dietary components that potentially influence CVD progression and prevention, this review attempted to provide the current state of evidence on healthy dietary choices for CVD. We focus on the effects of individual macronutrients, whole food products, and dietary patterns on the risks of CVD, and the data from population-based trials, observational studies, and meta-analyses are summarized. Unhealthy dietary habits, such as high intake of saturated fatty acids, sugar-sweetened beverages, red meat, and processed meat as well as high salt intake are associated with the increased risk of CVD. Conversely, increased consumption of plant-based components such as dietary fiber, nuts, fruits, and vegetables is shown to be effective in reducing CVD risk factors. The Mediterranean diet appears to be one of the most evidence-based dietary patterns beneficial for CVD prevention. However, there is still great debate regarding whether the supplementation of vitamins and minerals confers cardioprotective benefits. This review provides new insights into the role of dietary factors that are harmful or protective in CVD, which can be adopted for improved cardiovascular health.

## 1. Introduction

Cardiovascular diseases (CVDs), which include aortic atherosclerosis, coronary heart disease (CHD), stroke, and peripheral artery disease, are a group of disorders that affect heart and blood vessels. They represent a major public health concern and the leading cause of mortality worldwide [1]. According to the data from the American Heart Association (AHA), 26.1 million persons in the United States have some form of CVD, and over 800,000 people die of CVDs annually [2]. The annual direct and indirect costs of CVD deaths total more than USD 316.1 billion [2]. Globally, 17.9 million people die from CVD each year, according for about 32% of all deaths [3]. It is estimated that over 80% of deaths from CVD take place in low-income and middle-income countries [4]. Therefore, it is imperative to develop effective and affordable strategies for the prevention and treatment of CVD.

Age, gender, and genetic determinants are unmodifiable factors contributing to the development of CVD. Recently, much emphasis has been placed on the impact of modifiable factors. Poor dietary habits serve as one of the unhealthiest lifestyles for CVD morbidity and mortality (Figure 1) and account for about 10 million deaths worldwide [5]. Nutritional research in the context of CVD prevention has evolved in the past decades and shifted from single nutrients and specific foods to dietary patterns. However, the optimal components of a diet for improving cardiometabolic health remain uncertain. Hence, a comprehensive and systematic evaluation of the relation between diet and CVD is crucial for establishing guidelines of a high-quality diet and developing intervention strategies to reduce cardiovascular risk.

In this review, we performed an up-to-date summary of knowledge from population-based randomized control trials (RCTs), observational studies, and meta-analyses in an effort to comprehensively examine the role of specific diet components and dietary patterns in influencing CVD risk and health outcomes. We provided an evidence-based overview on healthy foods, dietary choices, and nutrition guidance for general health and CVD management.

## 2. Literature Search Strategy

The literature was searched and collected in PubMed databases. The key words “cardiovascular disease” and its combination with “dietary fat”, “dietary carbohydrate”, “dietary protein”, “vitamins and minerals”, “dietary fiber”, “sugar-sweetened beverages”, “red meat”, “processed meat”, “poultry”, “fish”, “nuts”, “fruits and vegetables”, “salt”, “sodium”, “Mediterranean diet”, “vegetarian diet”, “ultra-processed foods”, “ketogenic diet”, or “intermittent fasting” were searched. Further manual filtering of search results was carried out, and this article focuses mainly on the data included in RCTs, prospective cohort studies, and meta-analyses.

## 3. Macronutrients

### 3.1. Fat

Dietary fats mainly comprise triacylglycerols consisting of three individual fatty acids, each linked by an ester bond to a glycerol backbone. To prevent the development of CVD, reduced intake of dietary fats has been recommended by FAO and WHO. The dietary fats in foods can be classified into four major types: saturated fats, trans fats, monounsaturated fats, and polyunsaturated fats.

Saturated fats are made up of carbon atom chains that are all bonded with hydrogen. Reduction in dietary saturated fat intake is considered to be beneficial for cardiovascular health and is recommended by the current dietary guidelines. The scientific rationale for decreasing dietary saturated fat is based on the assumption that excess intake of dietary saturated fat may lead to increased levels of low-density lipoprotein cholesterol, which is a crucial risk factor for CVD progression. The association between saturated fat intake and CVD risk has been reported by both prospective cohort studies and RCTs (Table 1). Low intake of saturated fats appears to reduce the risk of cardiovascular events, and the effects may depend on the food source. Specifically, meat saturated fats are positively associated with CVD risk, whereas dairy saturated fats are inversely associated with CVD risk [6]. In the Prospective Urban Rural Epidemiology (PURE) study, the total fat and subtypes of fat including saturated and unsaturated fatty acids are found to be associated with low risk of total mortality and stroke, but are not associated with CVD mortality [7]. Moreover, the Japan Collaborative Cohort Study for Evaluation of Cancer Risk Study reports that saturated fatty acids intake is inversely associated with mortality from stroke [8]. The heterogeneity of the associations between fatty acid biomarkers and coronary risk is observed in another meta-analysis [9]. These findings show that saturated fatty acids intake had little or no effect on all-cause or cardiovascular mortality.

Likewise, there is also insufficient evidence supporting the positive associations between the intake of dietary saturated fats and the risk of CVD [10]. The results from the Women’s Health Initiative RCTs show that a low-fat diet does not have beneficial effects on CVD risk and total mortality in postmenopausal women [11]. Therefore, it is still uncertain whether a diet high in saturated fatty acids can play a causal role in contributing to CVD risk.

A prospective analysis of the PREvención con DIeta MEDiterránea (PREDIMED) study with 7038 participants at high CVD risk reports that saturated fatty acid and trans fat intake are associated with a high risk of CVD, whereas intake of monounsaturated fatty acids and polyunsaturated fatty acids are inversely associated with CVD death [12]. In line with these findings, a current dose–response meta-analysis of cohort studies shows that higher intake of dietary trans fatty acids is associated with a greater risk of CVD, whereas polyunsaturated fatty acids intake is inversely associated with CVD risk [13]. The Mediterranean diet, which is high in monounsaturated fatty acids and polyunsaturated fatty acids, and low in saturated fatty acids and trans fatty acids, is observed to effectively prevent the risk of major cardiovascular events [14]. Specifically, increased intake of linoleic acid, the n-6 polyunsaturated fat primarily from vegetable oils and nuts, is associated with a low risk of both total CHD events and deaths [15]. Therefore, replacing saturated and trans unsaturated fats with monounsaturated and polyunsaturated fats may be recommended to reduce the occurrence of clinical CHD events (Figure 2A) [16].

### 3.2. Carbohydrate

Carbohydrates are the main source of energy intake in diet and play a major role in health outcomes. Given the potential association between dietary fat intake and CVD risk, replacement of fats with carbohydrates is considered to be beneficial in the prevention of CVD. However, the findings regarding the association of dietary carbohydrate intake with CVD events are inconsistent. 

Results from previous trials and cohort studies have reported that there is an association between low carbohydrate intake and reduced mortality (Table 1) [17,18]. For instance, a low-carbohydrate, high-protein, high-fat diet is found to be beneficial in reducing body weight and improving risk factors for CHD in obese persons [19,20]. In a meta-analysis involving 11 RCTs with 1369 participants, the subjects on a low-carbohydrate diet had greater weight loss but increased cholesterol compared to those on a low-fat diet [21]. Another prospective cohort study showed that a moderate carbohydrate intake is associated with reduced risk of heart disease and stroke [22]. Glycemic control is a vital approach for CVD prevention. Studies have suggested that carbohydrate-restricted diets have beneficial effects on glycemic control. In a recent RCT, a non-calorie-restricted low-carbohydrate diet improved glycemic control and abdominal adiposity, and did not adversely affect cardiovascular risk factors in patients with type 2 diabetes [23]. Thus, these findings show that reducing carbohydrate intake appears to be an effective nutritional strategy to reduce cardiovascular risk factors. However, the results of these studies cannot explain whether the benefits of dietary intervention are attributable to the low carbohydrate intake, high protein (plant-based or animal-based) intake, or energy intake reduction.

In a large prospective cohort study of 135,335 individuals from 18 countries, a high intake of carbohydrates was found to be associated with an increased risk of total mortality [7]. A recent dose–response meta-analysis of 19 cohort studies including 15,663,111 participants showed that higher carbohydrate intake is associated with a slight increase in CVD risk in women, but no association is found in men [24]. Interestingly, another meta-analysis of prospective cohort studies showed that both high and low carbohydrate intake are associated with increased mortality, reflecting a potential U-shaped relationship between carbohydrate intake and mortality [25]. The data from the UK Biobank cohort show that distinct carbohydrates display different associations with mortality and CVD risk, indicating that the sources and types of carbohydrates, such as sugar, starch, and fiber, but not the total carbohydrates, should be taken into account [26]. 

Glycemic index (GI) is a value assigned to a specific food by measuring the incremental area under the blood glucose response curve after consuming the food. The quantity and quality of carbohydrates affect the GI, and a high GI is considered to be a feature of poor-quality carbohydrate foods. A previous meta-analysis of prospective cohort studies showed that diets with high GI and load are associated with CHD events in women but not in men [27]. A recent prospective study of 137,851 participants from diverse countries and regions showed that diets with a high GI are associated with a higher risk of CVD and death than diets with a low GI [28]. Thus, these studies suggest that the GI may be used as an indicator of carbohydrate quality related to CVD or other chronic diseases. Improving carbohydrate quality by lowering the GI is important for dietary intervention in preventing the adverse outcomes associated with CVD (Figure 2B). 

### 3.3. Protein

Dietary protein plays a critical role in promoting health. It acts on the metabolic targets involved in satiety, appetite, and energy metabolism. The early studies showed that consumption of high-protein diets may contribute to the prevention of obesity and metabolic syndrome and may provide benefits in reducing body weight, fat mass, and triglycerides (Table 1) [29,30]. Moreover, the data from prospective cohort studies have shown that replacing carbohydrates with protein is associated with a low risk of ischemic heart disease (IHD) and an improvement in cardiometabolic risk factors [31,32]. Notably, a prospective Nurses’ Health Study of U.S. women suggests that higher intake of red meat protein is significantly associated with elevated risk of CHD, while increased intake of poultry and fish protein is significantly associated with lower risk of CHD [33]. 

Dietary proteins can be classified on the basis of their plant or animal origin, and varied protein food sources may cause different effects on health outcomes. In the Rotterdam Study of a population-based cohort in the Netherlands, higher total protein intake was found to be associated with higher all-cause mortality, and this is driven mainly by higher animal protein intake and CVD mortality [34]. The data from two prospective US cohort studies show that animal protein intake is associated with higher risk of CVD mortality, whereas higher plant protein intake is associated with lower all-cause mortality [35]. Consistently, the findings from the Adventist Health Study-2 cohort study show that higher animal protein intake is associated with high CVD mortality, but plant protein intake is not associated with CVD mortality [36]. In a recent systematic review and dose–response meta-analysis of prospective cohort studies, the association between intake of total, animal, and plant protein and CVD risk was examined, and the results show that intake of plant protein, but not total or animal protein, is associated with low CVD risk and mortality [37]. Another prospective cohort study comprising 70,696 Japanese adults also showed that higher plant protein intake is associated with lower total and CVD mortality [38]. In line with these findings, small but significant associations between high intake of plant protein and low overall and CVD mortality are observed in a large US prospective cohort [39].

It should be noted that dietary protein is not consumed in isolation but as part of a whole food. The nonprotein nutrients and bioactive components in a food matrix may also influence CVD risk factors. Replacement of carbohydrate with plant-derived, but not animal-derived, protein or fat has been reported to be associated with lower mortality [25]. Moreover, replacement of red or processed meat protein with plant protein is associated with a decreased risk of CVD-related mortality [38].

Collectively, these findings prove that high plant-based protein intake contributes to long-term health outcomes and should be encouraged in the global dietary guidelines. Future studies are needed to unravel the underlying mechanisms responsible for the specific sources of plant-based protein on health. Moreover, the relative contribution of plant-based protein sources should be clarified in the current dietary guidelines (Figure 2C).

### 3.4. Vitamins and Minerals

Vitamins, including vitamins A, C, D, E, and K and the B vitamins, are chemically organic compounds essential for maintaining normal cellular metabolism. Minerals, including calcium, iron, and zinc, are important inorganic substances that maintain the normal function of human health. Vitamins and minerals are commonly used as dietary supplements because of their potential benefits in improving health and preventing diseases. In postmenopausal women, intake of vitamin E, but not vitamins A and C, has been reported to be inversely associated with the risk of CHD [40]. The preventive benefits of folic acid and B vitamins in stroke prevention are also reported [41]. However, multivitamins, vitamins C and D, β-carotene, calcium, and selenium do not exhibit positive effects in reducing CVD risk, and excessive doses of vitamin supplements may even cause certain adverse effects [42]. In a systematic review for the US Preventive Services Task Force, the benefits and harms of vitamin and mineral supplementation in healthy adults were evaluated, and the results show that supplementation with vitamin and minerals is associated with little or no benefit in preventing CVD [43,44]. Thus, there is still not enough evidence to fully support the benefits of vitamins and minerals for CVD prevention. 

Vitamin D is a fat-soluble vitamin that acts as a steroid hormone. The main function of vitamin D is to increase calcium absorption for bone mineralization. Previous cross-sectional studies have provided evidence supporting that vitamin D deficiency is associated with increased risk of CVD. For instance, the data from the National Health and Nutrition Examination Survey show that hypovitaminosis D is prevalent in US adults with certain CVDs such as CHD and heart failure [45]. Deficiency of 25-hydroxyvitamin D is found to be associated with high prevalence of myocardial infarction and heart failure [46]. The association between vitamin D deficiency and CVD risk has been evaluated in cohort and RCT studies. The Framingham Offspring Study reports that individuals with 25-hydroxyvitamin D < 15 ng/mL have a hazard ratio of 1.62 for developing cardiovascular events when compared with those with 25-hydroxyvitamin D > or =15 ng/mL, suggesting that vitamin D deficiency is associated with incident CVD [47]. Consistently, a nonlinear Mendelian randomization analysis supports a role for vitamin D deficiency in the risk of CVD [48]. However, recent RCT studies do not support a positive effect of vitamin D supplementation on reducing the CVD risk. In a RCT with participants from New Zealand, monthly high-dose vitamin D supplementation did not prevent CVD [49]. In a RCT of 25,871 healthy African Americans (The VITamin D and OmegA-3 TriaL), daily supplementation with high-dose vitamin D3 (2000 IU/day) for 5 years was not able to reduce the incidence of major cardiovascular events including myocardial infarction, stroke, and CVD mortality [50]. Consistently, in another RCT among Finnish elderly subjects, vitamin D3 supplementation with doses of 1600 IU/day or 3200 IU/day for 5 years was also not effective in preventing the incidences of major CVD events [51]. Thus, the available evidence does not support significant benefits or harms of vitamin D on cardiovascular risk (Figure 2D). 

### 3.5. Dietary Fiber

Dietary fiber is known as an important part of a healthy diet. It is defined as the edible parts of plants or analogous carbohydrate polymers that are neither digested nor absorbed in the small intestine. According to water solubility properties, dietary fiber can be classified into soluble or insoluble fiber. Soluble fiber includes β-glucan, fructans, pectin, and mucilage, and insoluble fiber includes cellulose, hemicellulose, lignin, and chitin. The regular consumption of fruits, vegetables, and cereals has been encouraged by dietary guidelines as important sources of dietary fiber. WHO states that a healthy diet should contain more than 25 g of dietary fiber per day. Intakes in the range of 25–30 g of fiber daily is recommended by most European countries [52]. Several RCTs have investigated the impact of fiber-enriched foods on CVD risk factors. For instance, daily consumption of 20 g of quinoa for four weeks has been reported to reduce CVD risk markers including blood cholesterol and blood glucose in overweight participants [53]. In another RCT study, consumption of barley β-glucan was effective in circulating cholesterol levels [54]. A short-term, prospective, open-labeled, randomized controlled, parallel group study found that daily intake of soluble fiber from oats exerts beneficial effects on lipid parameters in hypercholesterolemic Indians [55]. Pre-germinated brown rice bran extract containing acylated steryl glucosides has been reported to reduce the risk of atherosclerosis in postmenopausal Vietnamese women [56]. 

A previous meta-analysis of 10 cohort studies showed that there is an inverse relationship between intake of dietary fiber and the risk of CHD [57]. Similarly, the data from another meta-analysis of cohort studies show that a higher intake of dietary fiber, specifically from cereal or vegetable sources and rich in the insoluble type, is associated with a lower risk of both CVD and CHD [58]. The data from a recent meta-analysis of 185 prospective studies and 58 clinical trials has provided evidence showing that a high intake of dietary fiber is associated with a reduction in the risk of mortality and incidence of cardiometabolic events [59]. In another meta-analysis examining the association between fiber intake and CVD mortality, higher fiber intake was found to be associated with an improvement in cardiometabolic risk factors [60]. This study highlights the potential benefits of dietary fiber as adjunct therapy in subjects with CVD and hypertension. 

Taken together, based on the above findings from RCTs and observational studies, the available evidence strongly supports that intake of dietary fiber is associated with improved CVD risk factors. Further investigation is warranted to study the effects of specific types of dietary fiber on CVD prevention (Figure 2E).

**Table 1 nutrients-15-04898-t001:** Effects of macronutrients on cardiometabolic risk factors, cardiovascular disease, and mortality.

Macronutrients	Study	Research Object	MA (No. of Studies)	Exposure Measure	Follow-Up Period	Main Findings	Reference
Polyunsaturated fat and saturated fat	Meta-analysis of randomized controlled trials	Studies that randomized adults to increased total or n-6 PUFA consumption for at least 1 year and reported incidence of CHD.	YES (8)			Consuming PUFA in place of SFA can reduce the occurrence of clinical CHD events.	[16]
Saturated fat	Meta-analysis of prospective cohort studies	Studies of CHD, stroke, and cardiovascular disease of 347,747 participants.	YES (21)		5–23 years	No significant evidence supports associations between the intake of dietary saturated fat and the risk of CHD or CVD.	[10]
Saturated fat	The Japan Collaborative Cohort Study	58,453 Japanese men and women aged 40–79	No	FFQ	14.1 years	Saturated fatty acid intake is inversely associated with mortality from total stroke.	[8]
Saturated fat	The Multi-Ethnic Study	Participants who were 45–84 y old at baseline (*n* = 5209).	No	FFQ	10 years	Meat saturated fats are positively associated with CVD risk, whereas dairy saturated fats are inversely associated with CVD risk.	[6]
Linoleic acid	Meta-analysis of prospective cohort studies	Studies of linoleic acid and CHD with 310,602 participants.	YES (13)		5.3–30 years	Increased intake of dietary linoleic acid is associated with a low risk of CHD in a dose–response manner.	[15]
Fatty acids	Meta-analysis of observational studies	Studies of fatty acids, unsaturated fatty acids, and coronary disease of 659,298 participants.	YES (76)			There are no significant associations in prospective studies of coronary disease that involved assessment of dietary intake of long-chain w-3 and w-6 polyunsaturated fatty acids.	[9]
Dietary fat	The PREvención con DIeta MEDiterránea (PREDIMED) study	7038 participants with men (aged 55–80 y) and women (aged 60–80 y) at high CVD risk	NO	FFQ	6 years	Saturated fatty acid and trans fat intake are associated with a high risk of CVD, whereas MUFA and PUFA intake are inversely associated with CVD death.	[12]
Low-fat dietary pattern	The Women’s Health Initiative randomized controlled trial	Participants comprised 48,835 postmenopausal women aged 50–79 y.	NO	FFQ and performing laboratory analysis of blood specimens	8.3 years	Low-fat diets do not have beneficial effects on CVD risk and total mortality.	[11]
Fat	Prospective Urban Rural Epidemiology (PURE) study	A study of 135,335 individuals aged 35–70 years in 18 countries from five continents.	NO	FFQ	7.4 years	Total fats are associated with low risk of total mortality and stroke, but not with the risk of CVD, myocardial infarction, or cardiovascular disease mortality.	[7]
A Mediterranean diet supplemented with extra-Virgin olive oil or nuts	The PREDIMED study	A multicenter trial in Spain with 7447 participants (55 to 80 years of age, 57% women) who were at high cardiovascular risk.	NO	FFQ	4.8 years	The Mediterranean diet effectively prevents the risk of major cardiovascular events.	[14]
Dietary total fat	A dose–response meta-analysis of cohort studies	Studies of cohort studies reporting associations of dietary fat intake and risk of CVDs.	YES (43)			Total fat, SFA, MUFA, and PUFA intake are not associated with the risk of cardiovascular disease.	[13]
Low-carbohydrate diet	Randomized trial	A multicenter, controlled trial involving 63 obese men and women.	NO	participants met with registered dietitian to review dietary issues	1 years	In the first six months, the low-carbohydrate diet produced a greater weight loss compared with the conventional diet, but the differences are not significant at one year.	[19]
Low carbohydrate–high protein diet	A population-based prospective study	A cohort of 42,237 Swedish women (30–49 years old at baseline)	NO	FFQ	12 years	Low carbohydrate–high protein diet is associated with increased total and particularly cardiovascular mortality among women.	[17]
Low in carbohydrate diet	Randomized trial	Participants with 311 free-living, overweight nondiabetic, premenopausal women	NO	Received weekly instruction for 2 months, then an additional 10-month follow-up	1 years	Low-Carbohydrate diet benefit to weight loss and metabolic effects outcomes.	[20]
Low-carbohydrate diets	A prospective cohort study	Nurses’ Health Study and Health Professionals’ Follow-up Study with 85,168 women (aged 34–59 years at baseline) and 44,548 men (aged 40–75 years at baseline) without heart disease, cancer, or diabetes.	NO	FFQ	20–26 years	A low-carbohydrate diet based on animal sources is associated with higher all-cause mortality, whereas a vegetable-based low-carbohydrate diet is associated with lower cardiovascular disease mortality rates.	[18]
Diet with high glycemic index and load	Meta-analysis of prospective cohort studies	Studies showed associations of glycemic index and glycemic load with incidence of CHD including 240,936 participants.	YES (10)		6–25 years	Diet with high glycemic index and glycemic load diets are significantly associated with CHD events in women but not in men.	[27]
Low-carbohydrate diets	Meta-analysis of randomized controlled trials	Studies showed associations of low-carbohydrate diet, low-fat diet, weight loss, and cardiovascular disease with 1369 participants.	YES (11)		6–24 months	Low-carbohydrate diets have greater weight loss but increased LDL cholesterol.	[21]
Carbohydrate	Prospective Urban Rural Epidemiology (PURE) study	A large, epidemiological cohort study of 135,335 individuals aged 35–70 years	NO	FFQ	7.4 years	High intake of carbohydrate is associated with an increased risk of total mortality but is not associated with the risk of CVD or CVD mortality.	[7]
Carbohydrate	A prospective cohort study and meta-analysis	15,428 adults aged 45–64 years, in four US communities	YES (7)		25 years	Both high and low carbohydrate intake are associated with increased mortality.	[25]
Carbohydrate	Prospective population-based study of UK Biobank participants	The UK Biobank cohort of general population with 195,658 participants	NO	24-h recall	3 years	There were nonlinear associations between macronutrient intakes and health (mortality and CVD risk).	[26]
Diet with a high glycemic index	Prospective study on five continents	The study included 137,851 participants between the ages of 35 and 70 years living on five continents.	NO	FFQ	9.5 years	Diet with high glycemic index is associated with an increased risk of CVD and CVD mortality.	[28]
Carbohydrate and saturated fat	Prospective cohort study	9899 women (aged 50–55 years) were recruited into the Australian Longitudinal Study on Women’s Health.	NO		15 years	A moderate carbohydrate intake is associated with reduced risk of heart disease and stroke.	[22]
Low-carbohydrate diet	An open-label randomized controlled trial in Denmark	Study included 73 patients older than 18 years with type 2 diabetes.	NO	visits and telephone call	6 months	A non-calorie-restricted low-carbohydrate diet high in fat is significantly beneficial for glycemic control and body composition, without adversely affecting CVD risk factors in patients with T2D.	[23]
Carbohydrate	A dose–response meta-analysis	Studies about the relationship between dietary carbohydrate and the incidence of cardiovascular events and mortality.	YES (19)			Higher carbohydrate intake is associated with a slight increase in CVD risk in women but no association is found in men.	[24]
Dietary protein	A prospective cohort study	In the Nurses’ Health Study cohort of 80,082 women aged 34–59 y and without a previous diagnosis of ischemic heart disease, stroke, cancer, hypercholesterolemia, or diabetes.	NO	FFQ	14 years	Replacing carbohydrates with protein may be associated with a lower risk of ischemic heart disease.	[31]
Dietary protein	A prospective cohort study	84,136 women aged 30–55 years in the Nurses’ Health Study with no known cancer, diabetes, angina, myocardial infarction, stroke, or other cardiovascular disease	NO	FFQ	26 years	Higher intake of red meat and high-fat dairy are significantly associated with elevated risk of CHD, and CHD risk may be reduced by replacing sources of protein.	[33]
Energy-restricted high-protein, low-fat diet	Meta-analysis of randomized controlled trials	Studies that compared energy-restricted, isocaloric, high-protein, low-fat (HP) diets with standard-protein, low-fat (SP) diets included 1063 individuals.	YES (24)			Compared with an energy-restricted standard-protein, low-fat diet, an isocalorically prescribed high-protein, low-fat diet provided more benefits in reducing body weight, fat mass, and triglycerides.	[32]
Animal and plant protein	Two prospective US cohort studies	85,013 women and 46,329 men from the Nurses’ Health Study (1980–2012) and Health Professionals Follow-up Study (1986–2012)	NO	FFQ	32 years	Higher animal protein intake is positively, whereas plant protein is inversely, associated with all-cause mortality.	[35]
Plant and animal protein	The Adventist Health Study-2 cohort	81,337 men and women in the Adventist Health Study-2	NO	FFQ	9.4 years	Higher animal protein intake is associated with high CVD mortality, but no associations between plant protein intake and CVD mortality.	[36]
Animal and plant protein	A large prospective cohort study	Study included 70,696 participants in the Japan Public Health Center–based Prospective Cohort who were aged 45 to 74 years.	NO	FFQ	18 years	Higher plant protein intake is associated with lower total and CVD-related mortality, but animal protein intake is not associated with mortality outcomes.	[38]
Dietary protein	Meta-analysis of prospective cohort studies	Studies about associations of dietary protein from different sources with all-cause and cause-specific mortality with 350,452 participants.	YES (11)		12–28 years	Total protein intake is positively associated with all-cause mortality, driven mainly by a harmful association of animal protein with CVD mortality. Plant protein intake is inversely associated with all-cause and CVD mortality.	[34]
Plant and animal protein	A large prospective cohort study	Study included 416,104 men and women aged 50 to 71 in the US National Institutes of Health–AARP Diet and Health Study.	NO	FFQ	16 years	There are small but significant associations between high intake of plant protein and low overall and CVD mortality.	[39]
Total, animal, and plant proteins	Dose–response meta-analysis of prospective cohort studies	Studies of the dose–response relation between intake of total, animal, and plant protein and the risk of mortality from all causes, cardiovascular disease, and cancer.	YES (32)		3.5–32 years	Intake of plant protein is associated with low CVD mortality risk.	[37]
Dietary antioxidant vitamins	A prospective study	34,486 postmenopausal women aged 55 to 69 years with no cardiovascular disease	NO	FFQ and 24-h recall	7 years	The intake of vitamin E, but not vitamins A and C, from food is inversely associated with the risk of death from CHD.	[40]
Vitamin D	Framingham Offspring Study	1739 participants (mean age 59 years; 55% women; all white)	NO	FFQ		Deficiency of vitamin D is associated with incident cardiovascular disease.	[47]
Vitamin D	Cross-sectional study	The data from the National Health and Nutrition Examination Survey (NHANES) with 8351 participants	NO			Vitamin D deficiency is associated with increased risk of CVD.	[45]
Vitamin D	Cross-sectional analysis	The data from the Third National Health and Nutrition Examination Survey (1988–1994) with 16,603 men and women aged 18 years or older	NO			25-hydroxyvitamin D deficiency is found to be associated with high prevalence of angina, myocardial infarction, and heart failure.	[46]
Vitamin D	A randomized clinical trial	The study recruited participants mostly from family practices in Auckland, New Zealand, with 5110 participants aged 50 to 84 years.	NO	questionnaire	3.3 years	Monthly high-dose vitamin D supplementation does not prevent CVD.	[49]
Supplemental vitamins and minerals	Meta-analyses of randomized controlled trials	Studies of dietary supplements and cardiovascular disease outcomes and all-cause mortality	YES (179)			Folic acid and B vitamins had preventive benefits for stroke.	[41]
Vitamin D	Randomized trials	Among men 50 years of age or older and women 55 years of age or older in the United States	NO	FFQ	5.3 years	Supplementation with vitamin D does not result in a lower incidence of invasive cancer or cardiovascular events compared with placebo.	[50]
Supplemental vitamins and minerals	Meta-analyses of randomized controlled trials	Studies of dietary supplements and cardiovascular disease outcomes and all-cause mortality	YES (35)			Niacin shows an increased risk of all-cause mortality. However, multivitamins, vitamins C and D, β-carotene, calcium, and selenium do not exhibit positive effects in reducing the risk of CVD.	[28]
Vitamin and mineral supplements	Pooled analyses of RCTs and observational cohort studies	RCTs of vitamin or mineral use among adults without cardiovascular disease or cancer and with no known vitamin or mineral deficiencies; observational cohort studies examining serious harms.	YES (84)			Supplementation with vitamins and minerals is associated with little or no benefit in preventing cancer, CVD, and death.	[43]
vitamin D	Non-linear Mendelian randomization analyses	The analysis was conducted in the UK Biobank with 44 519 CVD cases and 251,269 controls.	NO			Vitamin D deficiency can increase the risk of CVD.	[48]
Vitamin D	A randomized controlled trial	The randomized, placebo-controlled trial among 2495 male participants ≥60 years and postmenopausal female participants ≥65 years from a general Finnish population who were free of prior CVD or cancer.	NO	Annual study questionnaires and national registry data	5 years	Supplementation with vitamin D3 does not lower the incidences of major CVD events or invasive cancer.	[51]
Dietary fiber	A pooled analysis of cohort studies	Studies about the association between dietary fiber intake and the risk of coronary heart disease	YES (10)			There is an inverse relationship between intake of dietary fiber and the risk of CHD.	[57]
Dietary fibre	Meta-analysis	Studies about the association of dietary fiber and cardiovascular or coronary heart disease	YES (19)		≥3 years	A higher intake of dietary fiber is associated with a lower risk of both CVD and CHD.	[58]
Rice bran extract	Randomized controlled trial	Single-blind design study with 60 postmenopausal Vietnamese women (45–65 y old) with high LDL cholesterol levels (over 140 mg/dL)	NO	questionnaires	6 months	Pre-germinated brown rice bran extract containing acylated steryl glucosides associated with the reduction in the risk of atherosclerosis.	[56]
Barley β-glucan	Randomized controlled trial	Crossover study with mild hypercholesterolemia participants (*n* = 45)	NO		5 weeks	Consumption of barley β-glucan is found to be effective in circulating cholesterol levels.	[54]
Quinoa	Randomized controlled trial	Crossover designed study with 37 healthy overweight men (35–70 years) completed a 4-week crossover intervention.	NO	FFQ	6 months	Daily consumption of 20 g quinoa can reduce CVD risk markers including blood cholesterol and blood glucose in overweight participants.	[53]
Dietary fiber	Meta-analyses	Studies about indicators of carbohydrate quality and noncommunicable disease incidence, mortality, and risk factors	YES (243)			High intake of dietary fiber is associated with lower risk of mortality and incidence of cardiometabolic events.	[59]
Dietary fiber	Meta-analyses	Studies about dietary fiber in hypertension and cardiovascular disease	YES (15)			Higher fiber intake is shown to be associated with an improvement in cardiometabolic risk factors.	[60]

Abbreviations: SFA, saturated fatty acids; CHD, coronary heart disease; PUFA, polyunsaturated fatty acids; MUFA, monounsaturated fats.

## 4. Foods and Food Products

### 4.1. Sugar-Sweetened Beverages (SSBs)

SSBs, which include carbonated and noncarbonated soft drinks, fruit drinks, and sports drinks that contain added caloric sweeteners, are the largest source of added sugar in the diet in high-income countries. Given the emerging association of added sugars with cardiometabolic risk factors, health authorities including WHO and Dietary Guidelines for Americans have suggested that the consumption of SSBs should be reduced to <10% of total energy intake. SSBs consumption may increase CVD risk factors among American populations, especially among US children [61,62]. The data from the Coronary Artery Risk Development in Young Adults (CARDIA) study show that higher SSB consumption is associated with a range of cardiometabolic outcomes such as high low-density lipoprotein cholesterol, high triglycerides, and hypertension [63]. Consumption of SSBs has also been reported to be associated with increased risk of CHD and adverse changes in blood pressure, inflammatory factors, and leptin [64,65]. Moreover, the findings from cross-sectional studies and prospective analysis of two large US cohorts (Nurses’ Health Study and Health Professionals Follow-up Study) have shown that habitual intake of SSBs is positively associated with a greater risk of CVD mortality and adverse levels of multiple cardiometabolic biomarkers [66,67,68]. Of note, the intake of SSBs is rising in low- and middle-income countries due to urbanization and beverage marketing [69]. Although no significant association between CVD mortality and SSB intake is found in a Chinese adult cohort in Singapore [70], a positive association of SSB intake (≥2 servings/day) with CVD mortality is observed in a Chinese younger adult cohort [71]. Thus, reducing intake of SSBs is important to improve overall diet quality and cardiometabolic health. The public nutrition policies and regulatory strategies should continue to call for reduction in intake of SSBs on a global scale.

Artificially sweetened beverages (ASBs) have been suggested as an alternative to SSBs because they provide few calories but retain a sweet flavor. Although a high level of ASB intake is reported to have a slight positive association with CVD mortality, replacing SSBs with ASBs may be recommended to improve health and longevity [68]. A recent network meta-analysis of 17 RCTs suggests that using ASBs as a substitute for SSBs is associated with reduced body weight and cardiometabolic risk factors such as body mass index (BMI), body fat, and lipid percentage [72]. These findings provide evidence on the benefits of using ASBs as a replacement strategy for SSBs. However, controversial results are also reported in some studies. For instance, a direct association between high artificial sweetener intake and increased CVD risk has been established, suggesting that ASBs may not be a healthy substitute for SSBs [73]. A previous meta-analysis of seven prospective cohort studies with 308,420 participants showed that there is an association between consumption of ASBs and cardiovascular risk [74]. Consistently, results from a cohort of French NutriNet-Santé study including 104,760 participants show that a higher intake of sugary drinks and ASBs is associated with a higher risk of CVD [75]. Moreover, another meta-analysis of prospective cohort studies provides evidence that the habitual consumption of both SSBs and low-calorie sweetened beverages is associated with a high risk of CVD incidence and CVD mortality [76].

Accordingly, the debate on whether ASBs can be used as an alternative to SSBs is still ongoing. Future studies are needed to better understand the underlying mechanisms of the potential metabolic impact of ASBs on health. For policymakers to make recommendations about SSBs and cardiometabolic health, new evidence alongside existing the literature needs to be considered. Moreover, the interest in suitable alternative beverages for SSBs such as 100% fruit juice, water, tea, and coffee is growing, and their health effects over the life-course and CVD prevention need to be examined in future studies (Figure 3A).

### 4.2. Red Meat and Processed Meat

Meat consumption is considered a risk factor for cardiovascular and metabolic diseases. Unprocessed red meat and processed meat are important components of the US diet, and their consumption has been linked to the risk of CVD mortality in the US population [77,78]. According to WHO, red meat consumption is “probably carcinogenic” to humans, whereas processed meat is considered “carcinogenic” to humans [79]. The dietary patterns that are low in red and processed meat intake have been recommended by WHO and US Dietary Guidelines for Americans.

Accumulating evidence from observational studies has shown that red and processed meat intake is positively associated with the risk of developing CVD. For instance, a meta-analysis of 17 prospective cohort studies provides evidence showing that high consumption of total red meat and processed meat is associated with an increased risk of CVD mortality [77]. The data from the Isfahan Cohort Study show that red meat and red plus processed meat intake are positively linked with CVD mortality, but inversely associated with stroke risk [80]. In a large prospective Health Professionals Follow-up Study cohort, a greater intake of total, processed, and unprocessed red meat is found to be associated with a higher risk of CHD [81]. Notably, substituting high-quality plant-based protein foods for red meat is associated with a lower risk of CHD [81].

The evidence supporting the association of reduced intake of red meat and processed meat with decreased CVD risk is insufficient. A meta-analysis with 6,035,051 participants showed that there is very-low-certainty evidence supporting that dietary patterns with less red and processed meat intake may result in decreased risk for CVD [82]. In another meta-analysis of prospective cohort studies, a small reduced risk for CVD mortality, stroke, and myocardial infarction was found to be associated with a reduced intake of processed meat and unprocessed red meat [83]. Consistently, a systematic review of randomized trials showed that diets lower in red meat may have little or no effect on all-cause mortality and nonfatal CVD [84].

Another point that should be emphasized is that red meat and processed meat may influence CVD risk differently. For example, in a meta-analysis including 17 prospective cohorts and 3 case-control studies with 1,218,380 individuals from 10 countries, intake of processed meat was found to be associated with high incidence of CHD, but no significant association was observed between the consumption of unprocessed red meat and total meat and the risk of CHD [85]. A recent large multinational prospective study showed that a higher intake of processed meat is associated with a higher risk of mortality and major CVD, but no significant association between unprocessed red meat and major CVD was observed [86]. These findings suggest that the effect of processed meat consumption on adverse cardiometabolic outcomes is somewhat greater than that observed for unprocessed red meat. Further studies focusing on revealing the mechanisms of how unprocessed red and processed meat consumption influence risk of cardiometabolic diseases are urgently needed for dietary and policy recommendations (Figure 3B).

### 4.3. Poultry and Fish

Although the positive associations between consumption of processed meat or unprocessed red meat and CVD risk have been established, the association of white meat intake with CVD-related incidents is still uncertain. White meat intake is considered to show healthier advantages compared to processed meat or unprocessed red meat. In the National Institutes of Health (NIH)-AARP Diet and Health Study, substituting white meat, particularly unprocessed white meat, is found to be associated with reduced risk of all-cause mortality [87]. A meta-analysis of prospective cohort studies reports that high intake of white meat is associated with reduced risk of stroke and stroke-related death [88]. Another meta-analysis of 13 cohort studies comprising 1,674,272 individuals show no significant association between intake of white meat and CVD mortality [89]. Moreover, a robust and inverse association between white meat consumption and all-cause mortality was observed in a recent meta-analysis of prospective cohort studies [90]. Collectively, these findings suggest that white meat may be a healthier and more sustainable alternative to red and processed meat consumption.

Poultry and fish are the major proportions of white meat consumption. Poultry meat is generally considered a healthy food because it provides high-quality protein and is often lower in fat than other animal meat. Moreover, poultry meat is reasonably affordable and accessible, with high rates of consumption globally. Fish has always been a main ingredient in the diet of populations living close to the sea. The diversity of the fish family provides quality protein, vitamins, essential fatty acids and micronutrients. Although evidence from RCTs is limited, several observational studies have been conducted to study the associations of poultry and fish consumption with the incidence of CVD. In a prospective cohort study of 29,682 US adults, consumption of poultry or fish was found to be not significantly associated with all-cause mortality, whereas intake of poultry but not fish is significantly associated with incidence of CHD, stroke, heart failure, and CVD [78]. The data from another UK Biobank prospective study show that eating fish rather than poultry is associated with a lower risk of adverse cardiovascular outcomes, including stroke, myocardial infarction, and heart failure [91]. These findings suggest that replacement of red and processed meats with fish but not poultry may be a better choice among patients at high risk for CVD.

Prospective cohort studies and RCTs have provided robust evidence showing a dose-dependent inverse relationship between fish consumption and heart failure incidence, cerebrovascular risk, and IHD risk as well as CHD death [92,93,94,95]. A large prospective study with 409,885 participants from nine European countries (European Prospective Investigation Into Cancer and Nutrition) reports that although there is no clear association between the consumption of white fish or fatty fish and IHD risk, a borderline significant inverse association is observed in the substitution analyses [96]. A pooled analysis of four prospective cohort studies including 191,558 people from 58 countries shows that a fish intake of 175 g (two servings of fish) weekly is reported to be associated with lower risk of major CVD events and total mortality among high-risk individuals or patients with existing vascular disease [97]. Consistent with these findings, the consumption of two servings of fish per week has been recommended by the American Heart Association and European Society of Cardiology (ESC). The beneficial effect of fish consumption on cardiovascular incidents is attributed mainly to omega-3 polyunsaturated fatty acids, such as eicosapentaenoic and docosahexaenoic acids [98,99]. Further studies are required to determine whether other nutrient ingredients in fish may also contribute to its cardioprotective effect on cardiovascular risk. Upcoming clinical trials are required to take more details into consideration (Figure 3C).

### 4.4. Nuts

Nuts, including tree nuts and peanuts, are unique plant food products that are rich in unsaturated fatty acids, protein, fiber, vitamins, minerals, and other bioactive compounds, such as phenolic antioxidants and phytosterols [100]. Due to their unique nutrient composition, nuts are thought to be beneficial for improving health.

There is compelling evidence showing that nut intake confers protection against CVD. The early prospective cohort investigation of the Adventist Health Study and the Physicians’ Health Study has reported that consumption of nuts is associated with a reduced risk of both fatal CHD, nonfatal myocardial infarction, and sudden cardiac death [101,102,103]. Soon thereafter, the inverse association of nut intake with death from heart disease such as IHD and CVD was observed in other large prospective cohort studies such as the Nurses’ Health Study and the Health Professionals Follow-up Study [104,105,106,107,108]. Supporting this, the prospective evaluation from three large cohorts including 71,764 US residents of African and European descent and 134,265 Chinese participants shows that nut consumption is associated with a reduced risk of CVD mortality in subjects with low socioeconomic status [109], highlighting that intake of nuts appears to be a cost-effective measure to improve cardiovascular health.

Studies have shown that a one-serving (28 g) increase in nut intake per day is associated with a 29% and 21% reduction in the relative risk of CHD and CVD, respectively, when compared with not eating nuts [110,111]. Moreover, frequent intake of nuts including tree nuts, peanuts, and walnuts has been associated with low CVD incidence and mortality [112,113]. Of note, the evidence regarding the association between nut intake and stroke is limited and still uncertain. Prospective cohort studies have provided evidence showing that high intake of dietary nuts is inversely associated with stroke risk [114,115]. However, no significant association between nut consumption and the risk of total or ischemic stroke is reported in the prospective cohort of 21,078 participants of the Physicians’ Health Study [116] and the prospective cohort of 26,285 participants of the European Prospective Investigation into the Cancer and Nutrition Potsdam Study [117].

Overall, these studies suggest that increasing nut consumption should be encouraged as a crucial part of a healthy dietary pattern to reduce the risk of CVD. Further research focusing on the role of specific nut types in influencing CVD outcomes particularly for stroke subtypes is needed (Figure 3D).

### 4.5. Fruits and Vegetables

Increased consumption of fruit and vegetables is known as the cornerstone of a healthy diet for CVD prevention. Over the past few decades, several epidemiological studies, including the first National Health and Nutrition Examination Survey Epidemiologic Follow-up Study [118] and the Atherosclerosis Risk in Communities study [119], have suggested that intake of fruit and vegetables may reduce CVD risk and may have beneficial effects on the total mortality and incidence of CHD [120]. However, there is still some inconsistency among people from different ethnic backgrounds. For instance, an inverse association between fruit and vegetable intake and CHD risk is found in Western populations, but not in Asian populations [121]. In a prospective cohort study (Prospective Urban Rural Epidemiology) including 135,335 participants from 18 low-income, middle-income, and high-income countries, higher fruit and vegetable consumption was found to be associated with a low risk of CVD, myocardial infarction, cardiovascular mortality, non-cardiovascular mortality, and total mortality in non-Western countries [122]. Moreover, a greater fruit rather than vegetable consumption has been found to be associated with a lower risk of CVD mortality in Chinese cohort studies [123,124]. In summary, these studies generally support a favorable relation between high amounts of fruit and vegetable consumption and reduced risk of CVD.

The dose–response relationship between fruit and vegetable consumption and CVD has been studied. A minimum of 400 g/day of fruits and vegetables has been recommended by many dietary guidelines. A fruit and vegetable intake over five servings/day has been reported to be associated with lower risk of CHD [125]. An intake of 800 g/day of combined apples/pears, citrus fruits, green leafy vegetables/salads, and cruciferous vegetables is able to reduce the risk of CVD [126]. The results from 2 prospective cohort studies (Nurses’ Health Study and Health Professionals Follow-up Study) and a meta-analysis of 26 prospective cohort studies show that consumption of approximately five servings of fruits and vegetables per day is associated with the lowest mortality, but higher intake does not reduce the additional risk [127]. However, an intake of three to four servings per day (equivalent to 375–500 g/day) of fruits and vegetables, which is affordable in low-income and lower-middle-income countries, is found to be as beneficial as higher amounts of intake in reducing total mortality [122]. Therefore, fruit and vegetable consumption is likely to be a cost-effective approach to prevent CVD in low-income countries.

Stroke is the third leading cause of death and a common cause of disability in developed countries. Early cohort studies have provided evidence supporting the inverse relation between fruit and vegetable intake and stroke [128]. For instance, the intake of cruciferous and green leafy vegetables and citrus fruit and juice displays protective effects on reducing ischemic stroke risk [129]. Similarly, a prospective cohort study of 54,506 Danish people showed that an increased intake of fruit reduces the risk of ischemic stroke [130]. A prospective cohort study of 40,349 Japanese showed that daily intake of green-yellow vegetables and fruits is associated with a lower risk of total stroke, intracerebral hemorrhage, and cerebral infarction mortality [131]. Individuals with more than five servings of fruit and vegetables per day are found to be associated with a lower risk of stroke when compared to those with fewer than three fruit and vegetable servings per day [132]. The data from a recent large cohort comprising 418,329 participants from nine European countries show that higher consumption of fruit and vegetables is inversely associated with the risk of total and ischemic stroke [133]. Thus, adherence to an increased intake of fruit and vegetables is recommended for the prevention of stroke (Figure 3E).

### 4.6. Salt and Sodium

Salt is an important nutrient component of a healthy diet, and the human body needs a certain amount of salt to maintain cellular homeostasis. A systematic analysis of 24 h urinary sodium excretion and a dietary survey worldwide reports that sodium intake has exceeded the recommended levels in almost all countries [134]. It is estimated that 1.65 million deaths from cardiovascular causes in 2010 are attributable to the consumption of more than 2.0 g of sodium per day [135]. Therefore, sodium reduction has been recommended by American and European guidelines for the prevention of hypertension and CVD occurrence [136,137]. The WHO has set a global target of reducing salt intake to an eventual target of <5 g daily in adults.

The role of sodium in cardiovascular health is to maintain intravascular volume. Accumulating evidence has shown that increased sodium intake is closely related to elevated blood pressure, an essential risk factor for CVD. In the INTERSALT study with 10,079 adults from 32 countries, a direct association between salt intake and blood pressure was observed [138]. Reduced salt intake has been reported to control blood pressure effectively [139]. A trial examining the effect of dietary sodium on blood pressure reported that a reduction in sodium intake to levels below the current recommendation of 2.3 g per day is able to lower blood pressure [140]. Consistently, a systematic review and meta-analysis of randomized trials shows that a longer-term modest reduction of salt intake from 9–12 to 5–6 g/day leads to falls in blood pressure and may reduce CVD [141]. Another meta-analysis of 14 cohort studies and 5 RCTs also shows a clear benefit of lower sodium intake on reduction in blood pressure [142].

Accumulating evidence has suggested that sodium intake is associated with an increased CVD risk. For instance, an early prospective study of 1173 Finnish people has reported that increased urinary sodium excretion can predict mortality and risk of CHD, suggesting that high sodium intake is a key risk factor for CVD [143]. A meta-analysis of 13 prospective studies comprising 170,000 people has reported that high salt intake is associated with a greater incidence of stroke and cardiovascular events [144]. Another meta-analysis of prospective studies also showed that higher sodium intake is associated with higher CVD mortality [145]. The results from long-term follow-up of two completed lifestyle intervention trials show that people with a sodium reduction have a lower risk of cardiovascular outcomes in 10 to 15 years after the trial [146]. A pooled analysis of data from four international prospective studies with 133,118 people from 49 countries shows that high sodium intake (greater than 6 g/day) is associated with an increased risk of CVD events and death in individuals with hypertension. Notably, low sodium intake (less than 3 g/day) is also associated with an increased risk of cardiovascular outcomes in individuals with or without hypertension, implying a J- or U-shaped association between salt intake and CVD risk [147]. In line with these findings, the data from clinical trials and observational studies have shown that reducing sodium intake to less than 5 g/day is effective in reducing blood pressure and CVD [148].

However, a few recent cohort studies show controversial results. For instance, a prospective cohort study reports that although the relationship between sodium intake and blood pressure is linear, no strong association of dietary salt intake with increased CVD risk is observed [149]. In a recent prospective cohort of 176,570 participants from the UK biobank, lower frequency of adding salt to foods was found to be associated with lower risk of CVD, particularly heart failure and IHD [150]. Collectively, although both low and high salt intakes are reported to be associated with increased risk of CVD, reducing salt intake at a modest level may be an effective and affordable strategy to prevent CVD. Future large and long-term trials are needed to confirm the effectiveness and safety of salt reduction for CVD events and mortality (Figure 3F).

### 4.7. Dairy Products

Dairy food products, which include butter, milk, cheese, and yogurt, are widely consumed worldwide and represent a major source of dietary saturated fatty acids. As stated above, a high intake of saturated fatty acids has been associated with an increased risk of CVD. This may imply that a reduction in dairy food consumption may cause decreased low-density lipoprotein cholesterol level in plasm and contribute to improved cardiovascular health. Circulating biomarkers of dairy fat provide objective measures of dairy fat intake. A systematic review and meta-analysis of prospective studies reports that no significant effects of circulating biomarkers of dairy fat on total CVD, CHD, or stroke, suggesting that higher dairy fat exposure is not associated with an increased risk of CVD [151]. Another systematic review and meta-analysis showed that higher levels of both odd-chain dairy fat biomarkers are associated with lower CVD risk [152]. Moreover, a large, multinational, prospective cohort study involving participants from 21 countries in five continents provided evidence showing higher dairy consumption is associated with lower risks of mortality and CVD, particularly stroke [153]. In fact, dairy foods are a heterogeneous group of products with different biochemical properties and nutritional composition. This means that the effect of dairy foods on cardiovascular health may depend on the specific food type. Indeed, the available evidence about the association of dairy foods consumption with CVD-related health outcomes is still controversial.

A previous meta-analysis of prospective cohort studies reported that there is an inverse association between dairy consumption and overall risk of CVD [154]. In line with these findings, dairy consumption was also found to be inversely associated with CVD, CHD, and stroke [155]. However, another meta-analysis of 29 prospective cohort studies with 938,465 participants showed that there is no association between dairy products and cardiovascular mortality [156]. In this study, the intake of total fermented dairy foods including cheese and yogurt was shown to be inversely associated with CVD risk, but no association was found for yogurt consumption [156]. The heterogeneity of individual dairy products was also observed in a systematic review and meta-analysis of cohort studies examining the association of dairy product intake with the risk of major atherosclerotic CVDs in the general adult population. The results show a positive association of high-fat milk and an inverse association of cheese with CHD risk [157]. Consistently, fermented dairy food intake is found to be associated with decreased CVD risk [158]. In the Kuopio Ischaemic Heart Disease Risk Factor Study, high intake of fermented dairy products showed an inverse association with the risk of CHD, whereas high intake of non-fermented dairy products was associated with increased risk of CHD [159]. Moreover, non-fermented milk intake was associated with an increased risk for developing cardiometabolic diseases [160]. In a recent meta-analysis of cohort studies with 896,871 participants, an inverse association between yogurt consumption and risk of all-cause and CVD mortality was found, whereas there was no significant association between yogurt consumption and risk of cancer mortality [161].

Overall, these findings suggest that fermented and non-fermented dairy products can have opposite associations with the risk of CVD. The effect of dairy food consumption on cardiovascular health may depend more on the food type relative to the fat content within the products. Further study is required to elucidate the mechanisms by which fermented dairy products improve cardiovascular health.

## 5. Dietary Patterns

### 5.1. Mediterranean Diet

The Mediterranean diet is one of the best-studied diets for cardiovascular health. It is a traditional dietary style around the Mediterranean Sea, and it is characterized by a relatively high proportion of plant foods such as fruits, vegetables, nuts, and cereals; poultry and fish consumed in low to moderate amounts; olive oil as the main source of fat; a low intake of dairy products, red meat, processed meats, and sweets; and wine in moderation [162]. It is noteworthy that the Mediterranean diet has variants depending on the characteristics of nutritional composition. The lack of a universal definition for the Mediterranean diet may lead to disparate conclusions across different studies. Despite this limitation, the Mediterranean diet is recommended by the American and the European societies for the management of cardiovascular risk.

Studies of intervention trials have provided mounting evidence to support the beneficial effects of the Mediterranean diet in reducing CVD morbidity and mortality. The results from the Lyon Diet Heart Study and the Seven-Countries Study have shown that a greater adherence to the Mediterranean diet confers protective effects in reducing the recurrence rate of myocardial infarction and the risks of incident CHD and stroke [163,164]. Compared to low-fat diets, Mediterranean diets are more effective in improving clinically relevant cardiovascular risk factors and inflammatory markers [165]. In a randomized trial involving persons at high cardiovascular risk (the PREDIMED trial), a Mediterranean diet supplemented with extra-virgin olive oil or nuts is found to be effective in reducing the incidence of major cardiovascular events when compared to a reduced-fat diet [14]. Furthermore, a recent large-scale, long-term clinical trial evaluating dietary pattern efficacy showed that the Mediterranean diet is superior to a low-fat diet in preventing secondary CVD recurrence [166]. Together, these findings support the beneficial effects of the Mediterranean diet for the primary and secondary prevention of CVD.

A number of observational studies have also provided evidence supporting the effectiveness of the Mediterranean diet in reducing CVD incidence and mortality. An umbrella review of meta-analyses on prospective cohort studies shows that a higher adherence to the Mediterranean diet is associated with a lower incidence and mortality of CVD [167]. In a Danish cohort study, the Mediterranean Diet Score was found to be inversely associated with cardiovascular incidence and mortality, but not with stroke incidence or mortality [168]. Similarly, although the Mediterranean diet is associated with a decreased risk of CVD, this protective effect only included ischemic stroke but not hemorrhagic stroke [169]. Moreover, the Mediterranean diet is able to exert a beneficial effect in reducing CVD and myocardial infarction incidence in individuals with diabetes [170,171].

Overall, the available evidence from clinical trials, observational studies, and meta-analyses strongly supports the Mediterranean diet as an ideal eating approach for cardiovascular health. The health benefits of the Mediterranean diet may be the synergistic result of multiple beneficial components of this dietary pattern but not a single component. This suggests that overall food patterns may represent the current effective dietary strategy for the management of CVD.

### 5.2. Vegetarian and Vegan Diets

A vegetarian diet is commonly defined as a plant-based dietary profile and is characterized by exclusion from consuming meat products such as beef, chicken, pork, and seafood [172]. Dairy products and eggs may be included in the vegetarian diet [172]. The vegetarian diet is the recommended dietary pattern that meets the AHA guidelines [173]. Moreover, a plant-based diet has been recommended by the EAT-Lancet Commission as a healthy diet for the prevention and treatment of CVD [174].

The studies from systemic reviews and meta-analyses have shown that a vegetarian diet is associated with reduced risk of CVD. For instance, in a meta-analysis evaluating the impact of a vegetarian diet on CVD mortality, nonvegetarians are reported to have a higher mortality from IHD than vegetarians [175]. The beneficial effect of a vegetarian diet on the incidence and mortality from IHD is also observed in another systematic review and meta-analysis of 98 cross-sectional studies and 10 cohort prospective studies with 130,000 vegetarian subjects [176]. Supporting this, another meta-analysis of eight studies with 183,321 participants provides evidence supporting the association between a vegetarian diet and a moderate cardiovascular benefit, but this association was driven mainly by a specific cohort (Seventh-Day Adventist) [177]. Moreover, a recent meta-analysis of 13 prospective cohort studies with 844,175 participants demonstrated that a vegetarian diet is associated with a reduced risk of CVD and IHD, but not stroke [178].

Studies have shown that the quality of plant-based diets is an important factor associated with the risk of CVD because not all plant foods are beneficial for health. Whole grains, fruits and vegetables, nuts and legumes, oils, tea, and coffee can be classified as healthy plant foods. Unhealthy plant foods mainly include juices and sweetened beverages, refined grains, potatoes and fries, and sweets. A high intake of healthy plant foods is reported to be associated with low CHD risk, whereas a high intake of unhealthy plant foods is associated with high CHD risk [179]. A higher intake of healthy plant foods is also associated with lower CVD mortality, whereas a higher intake of unhealthy plant foods is associated with a higher cardiometabolic disease risk [180]. For instance, potatoes are plant foods, but they have a high glycemic index and glycemic load. There are significant associations between potato intake and CVD risk factors among an Iranian population [181]. However, the evidence supporting the association with cardiovascular mortality is not available. Moreover, a recent study examining the association of a plant-based diet quality with the risk of stroke also provides evidence supporting the importance of adhering to a healthful plant-based diet in lowering the risk of total and ischemic stroke [182]. In a study of two prospective cohorts in Taiwan, a vegetarian diet was found to be associated with a lower risk of ischemic and hemorrhagic strokes [183]. Therefore, adherence to a diet containing plenty of healthy plant foods should be emphasized and recommended by nutrition and public health policies to improve cardiometabolic health outcomes in the future.

Compared to a vegetarian diet, a vegan diet is a dietary pattern characterized by the total elimination of all animal-origin products, including dairy and eggs. In a systematic review of studies evaluating the health impact of the vegan diet on the risks of primary, intermediate, and recurrent CVD, a vegan diet was found to be associated with a decreased risk of recurrent CVD events [184]. Notably, an increased risk of ischemic stroke is observed in vegans, and this may be due to the low intake of certain nutrients that are crucial for disease prevention [184]. In another study, a vegan diet was found to be associated with an 18% reduction in the relative risk of IHD, but no clear association was observed between vegan diet and CVD or stroke [178]. In a cross-sectional study, vegan children were found to have a healthier profile of cardiometabolic risk factors than omnivore children [185]. Interestingly, children consuming a vegan diet exhibit a low level of bone mineral content and height, and the restriction of animal-based foods in vegan children may contribute to nutritional deficiencies that cause abnormal development [185]. Thus, a vegan diet may be linked to micronutrient deficiency, and specific nutritional supplementation needs to be emphasized in future studies [186]. Overall, these findings highlight the importance of adhering to a well-balanced diet to prevent risk of CVD but, at the same time, avoid the risk of developing possible nutritional deficiencies.

### 5.3. Ultra-Processed Foods

According to the extent and purpose of industrial processing, foods and food products can be classified by the NOVA classification system into four groups: (1) unprocessed or minimally processed foods; (2) processed culinary ingredients; (3) processed foods; and (4) ultra-processed foods [187,188]. Ultra-processed foods are not “real food.” The concept of ultra-processed food was first proposed by researchers from the University of São Paulo [189]. They defined ultra-processed foods as “formulations of ingredients, mostly of exclusive industrial use, that result from a series of industrial processes” [187,188]. Examples of such foods include soft drinks, sweet or savory packaged snacks, reconstituted meat products, and pre-prepared frozen dishes.

During the past decades, the availability of ultra-processed foods has increased substantially worldwide, leading to global concerns about their consumption related to cardiometabolic health. Recent epidemiological studies have suggested that higher consumption of ultra-processed food is associated with increased risk of CVD. In a large prospective cohort study (NutriNet-Santé), higher consumption of ultra-processed food was found to be associated with an increased overall risk of cardiovascular, coronary heart, and cerebrovascular diseases [190]. In the prospective Framingham Offspring Cohort study, consumption of ultra-processed foods was associated with an increased risk of cardiovascular events and mortality [191]. Another dose–response meta-analysis of seven cohort studies also showed that consumption of ultra-processed food is associated with an elevated risk of CVD-cause mortality [192]. Consistently, the UK Biobank Cohort prospective study has shown that a higher proportion of ultra-processed food intake is associated with CVD and all-cause mortality [71]. The Moli-sani Study reports that a diet rich in ultra-processed foods is associated with increased hazards of all-cause and CVD mortality among individuals with prior cardiovascular events [193]. A significant association between ultra-processed food intake and mortality has been observed in a prospective analysis within the Third National Health and Nutrition Examination Study (NHANES III) [194]. Moreover, ultra-processed food consumption may be a modifiable risk factor for cardiovascular consequences in children and adolescents [195].

This association between ultra-processed food consumption and CVD risk may be attributable to the nutritional composition of products, food additives, and neoformed contaminants contained in ultra-processed food. Further research is warranted to clarify the specific contributions and biological mechanisms of these factors to understand the causal role of ultra-processed food in CVD. Moreover, limiting the proportion of ultra-processed food or increasing the consumption of unprocessed or minimally processed foods in the diet should be recommended and advised by official nutritional guidelines and public health authorities.

### 5.4. Ketogenic Diet

A ketogenic diet is a dietary pattern characterized by a reduction in carbohydrates and a relative increase in the proportion of proteins and fats [196]. It is known to be effective in treating epilepsy and neurodegenerative disorders. Furthermore, it has been widely used in weight loss programs for metabolic abnormalities. Recently, studies have investigated the potential use of ketogenic diets for the prevention and treatment of CVD. The ketogenic diet has been reported to be associated with significant decreases in BMI, abdominal circumference, systolic blood pressure, fasting plasma glucose, and glycated hemoglobin [197]. Compared with a conventional low-fat diet, ketogenic diets can induce a long-term improvement in cardiovascular risk factors such as body weight, diastolic blood pressure, triglycerides, and HDL cholesterol [198]. Moreover, a short-term ketogenic diet managed in the setting of a general practice is able to improve the anthropometric, hemodynamic, and metabolic parameters related to CVD risk [199]. However, another meta-analysis including 19 RCTs with 3209 participants shows that there is probably little or no difference in changes in cardiovascular risk factors when comparing the ketogenic diet to isoenergetic diets [200].

It should be noted that a ketogenic diet may have some adverse side effects and increase the risk of CVD due to the increased intake of dietary fats. For instance, a 3-week low-carbohydrate–high-fat diet causes metabolic inflexibility by increasing the levels of free fatty acids, total cholesterol, and low-density lipoproteins, which are associated with increased risk for CVD [201]. In a randomized controlled feeding trial, feeding healthy women a ketogenic low-carbohydrate–high-fat diet for 4 weeks caused a deleterious blood lipid profile, implying an increased cardiovascular risk [202]. Thus, there are conflicting data regarding the effect of ketogenic diets on cardiovascular health, despite some improvements in cardiovascular risk factors being observed. Further studies are needed to increase knowledge of potential mechanisms and to ensure the efficacy and safety of ketogenic diets in the long term.

### 5.5. Intermittent Fasting

Intermittent fasting is defined as a form of time-restricted eating [203]. The potential health benefits of intermittent fasting for aging and life span have been demonstrated in animals and humans [204]. Recent studies have provided evidence showing the beneficial effects of intermittent fasting on multiple indicators of cardiovascular health. For instance, an intermittent fasting regimen is reported to be an effective dietary therapy in improving CHD risk factors such as visceral fat and adipokines [205]. In healthy, nonobese individuals, sustained intermittent fasting is beneficial in improving cardiovascular and metabolic risk factors, including blood pressure and lipid profile [206]. A recent review of 11 meta-analyses comprising 130 RCTs reported the beneficial effect of intermittent fasting on obesity-related cardiometabolic outcomes [207].

Time-restricted eating is a specific intermittent fasting protocol with a fasting period of 16 h and ad libitum feeding for 8 h, and it has been shown to be beneficial in improving cardiovascular risk factors in healthy males [208]. In a non-RCT trial among obese women, time-restricted eating promoted weight loss, despite not leading to significant changes in blood biomarkers associated with metabolic and cardiovascular risk [209]. Moreover, eating early in the day with an 8-h eating window from 7:00 to 15:00 is found to be more effective for weight loss and improving diastolic blood pressure than eating over a window of 12 or more hours at 14 weeks [210]. Nevertheless, the data from the TREAT Randomized Clinical Trial show that time-restricted eating does not change any relevant metabolic markers [211].

“Two-day” intermittent fasting is described as “an eating pattern in which there are 2 nonconsecutive days of calorie restriction a week and an ad libitum diet the other 5 days.” In a RCT with adults with metabolic syndrome, 2-day intermittent fasting was found to be associated with the mitigation of cardiometabolic risk factors and improvement of gut microbiota homeostasis [212]. Another RCT study showed that alternate day fasting, an intermittent fasting regimen with strict 36 h periods without caloric intake followed by 12 h intervals with ad libitum food consumption, is an effective strategy with cardioprotective effects in nonobese subjects [213]. A prospective cohort study with an embedded RCT showed that alternate-day fasting improves cardiovascular parameters and CVD risk among nonobese adults within four weeks [214]. A systematic review and meta-analysis showed that alternate-day fasting causes significant reduction in total cholesterol [215]. However, in a RCT on lean, healthy adults, another type of alternate-day fasting with 24 h fasting and 150% energy intake on alternate days was shown to have no specific effects on systemic markers of cardiovascular health [216]. Collectively, observational and clinical trials have provided evidence supporting that intermittent fasting may have certain benefits for CVD.

## 6. Conclusions

A healthy diet is the cornerstone of human health and well-being. It is estimated that diet-related risks account for 52% of all CVD deaths worldwide according to the Global Burden of Disease Study [5]. Accordingly, changing diet may be an important modifiable and cost-effective approach to reduce the burden caused by CVD [217]. Recently, the evidence on the components of a healthy diet in cardiovascular health is growing. A clearer understanding of the relation between dietary components and CVD is important for establishing early dietary guidance to reduce CVD risk. In this context, evidence-based guidelines may play a critical part in informing healthy food choices to promote cardiometabolic health. In this review, we summarized the evidence documenting the association of dietary components including individual nutrients, food products, and dietary patterns with CVD risk and mortality.

Healthy dietary choice is defined by greater adherence to dietary patterns that include high intake of dietary components associated with reduced risk of CVD and low intake of dietary components associated with increased risk of CVD. Increased consumption of plant-based components such as dietary fiber, nuts, fruits, and vegetables has been shown to be effective in reducing CVD risk factors. Conversely, unhealthy dietary choices, such as high intake of saturated fatty acids, sugar-sweetened beverages, red meat, and processed meat as well as high salt intake are associated with the increased risk of CVD.

We have summarized existing dietary recommendations from two leading international cardiac science societies ESC [218] and AHA [219] for CVD prevention (Table 2). The core information has considerable overlap and consistency with the evidence reviewed here. Both guidelines emphasize consumption of plant-based components such as fruits, vegetables, whole grains, and nuts. For animal-based foods, they both recommend increasing the consumption of fish. In addition, they all suggest reducing salt intake and SSBs, and they recommend limiting intake of alcoholic beverages. However, there are differences between the two guidelines. The AHA emphasizes the types of recommended foods, whereas the ESC pays more attention to nutritional considerations and provides detailed guidance on the quantity of recommended foods.

A diet with an increased intake of plant-based nutrients and foods including dietary fiber, nuts, fruits, and vegetables and a low intake of salt, saturated fatty acids, sugar-sweetened beverages, red meat, and processed meat is linked to a reduced risk of CVD. Furthermore, plant-based fats and proteins may be a key factor in the primary and secondary preventions of CVD. Notably, the evidence supporting the cardioprotective benefits of vitamin and mineral supplementation is insufficient. With regard to meat consumption, white meat, particularly fish, but not red meat and processed meat may be a better choice for a high-quality protein source. Accordingly, adhering to a balanced, diversified plant-based diet is a long-term approach to promote cardiovascular health (Figure 4).

Of note, nutrients are not consumed in isolation but as part of a food matrix. Therefore, it is difficult to control for the potential effects of other nutrients provided by a food. Recently, the focus of nutritional research for CVD has shifted from single nutrients and specific foods to dietary patterns. The study of whole food patterns may be more important and more interpretable than analyses focused on single nutrients or foods. This may be partially attributable to the fact that many associations between individual components of macronutrients and CVD risk are nonlinear, leading to confusing and conflicting findings that are not consistent with current dietary recommendations. Although there is an emerging association between ultra-processed food consumption and CVD risk, the Mediterranean diet appears to be the most evidence-based dietary pattern that is beneficial for CVD prevention. Further studies are warranted to clarify the mechanisms by which the specific factors of ultra-processed foods and Mediterranean diets affect cardiovascular health, and thereby help to establish reasonable guidelines of a high-quality healthy diet to reduce CVD risk.

Although the present article discusses healthy dietary choices for CVD, adopting such diets in real-world settings is still challenging. Factors such as cultural and geo-graphical influences, economic barriers, access to healthy foods, and the importance of early dietary intervention from a young age are all critical considerations. In fact, although health motivations may help compliance in the short term, altering entrenched dietary practices long term is difficult because diet not only holds nutritional significance but also carries social and cultural meanings. It is also important to realize that individuals with a lower socioeconomic status may encounter financial barriers to embracing a healthier and more sustainable diet. Moreover, changing dietary habits is not only an individual choice, but a challenge for the whole society. Implementing policy measures, such as advertising bans, taxation on unhealthy foods, or subsidies for nutritional options, will help people to make more healthy choices. The food industry, through producing foods that are not only healthy but also gastronomically appealing, may play a pivotal role. People have to be educated and supported in changing their lifestyle, so that healthy dietary choices will become easier to be implemented practically to reduce the burden of CVD.

Moreover, the current state of evidence showing the role of diet in affecting CVD risk is mainly from observational studies and meta-analyses, which can only provide an association analysis examining the relationships between dietary factors and CVD-related outcomes. The uncertainty of inherent methodological problems that are used to calculate diet quality may lead to inconsistent results. Limited evidence is available from a few clinical trials examining the effect of improved diet quality on CVD incidence or mortality, whereas there are still multiple confounding factors and possible bias about data collections. Therefore, future RCTs with a large sample size of participants across different countries and continents are required to confirm the findings from perspective cohort studies. In summary, this review provides evidence-based recommendations for making healthy dietary choices to promote cardiovascular health and reduce CVD risk.

## Figures and Tables

**Figure 1 nutrients-15-04898-f001:**
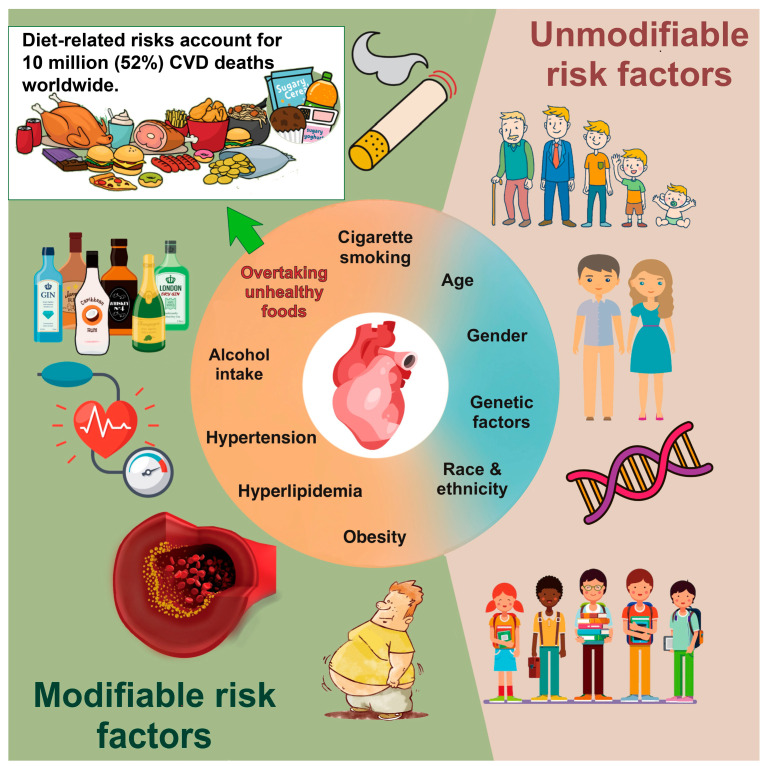
Factors contributing to the development of CVD.

**Figure 2 nutrients-15-04898-f002:**
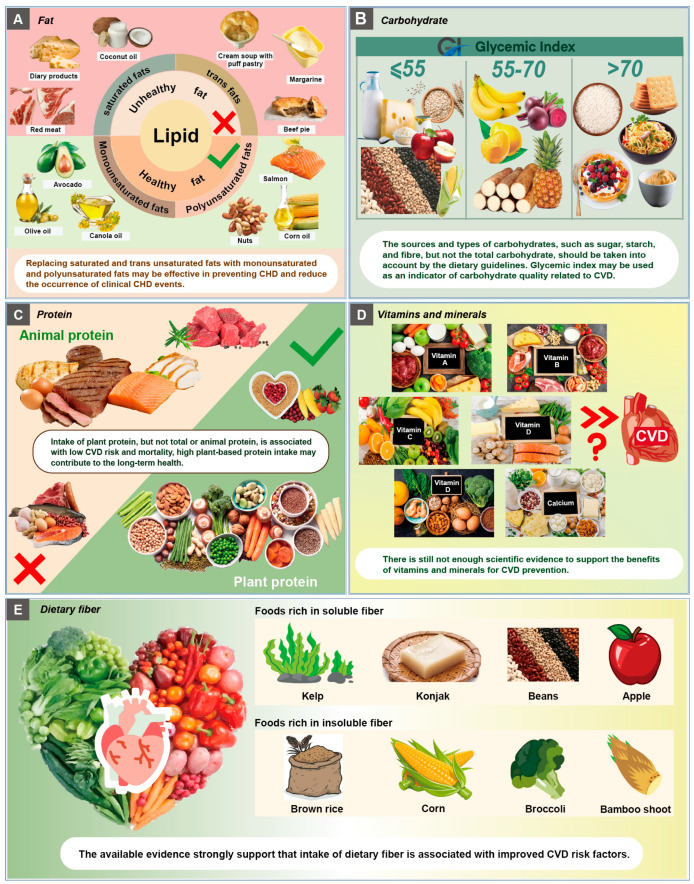
Suggested individual nutrient consumption according to the evidence on the association between dietary choices and cardiovascular disease management. (**A**) Fat; (**B**) Carbohydrate; (**C**) Protein; (**D**) Vitamins and minerals; (**E**) Dietary fiber.

**Figure 3 nutrients-15-04898-f003:**
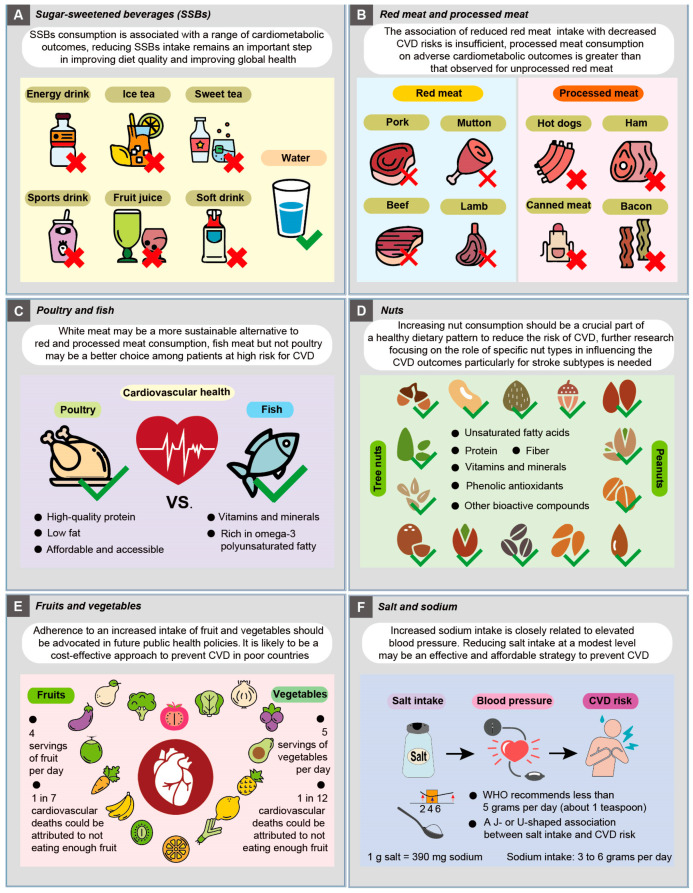
Suggested specific food consumption according to the evidence on the association between dietary choices and cardiovascular disease management. (**A**) Sugar-sweetened beverages; (**B**) Red meat and processed meat; (**C**) Poultry and fish; (**D**) Nuts; (**E**) Fruits and vegetables; (**F**) Salt and sodium.

**Figure 4 nutrients-15-04898-f004:**
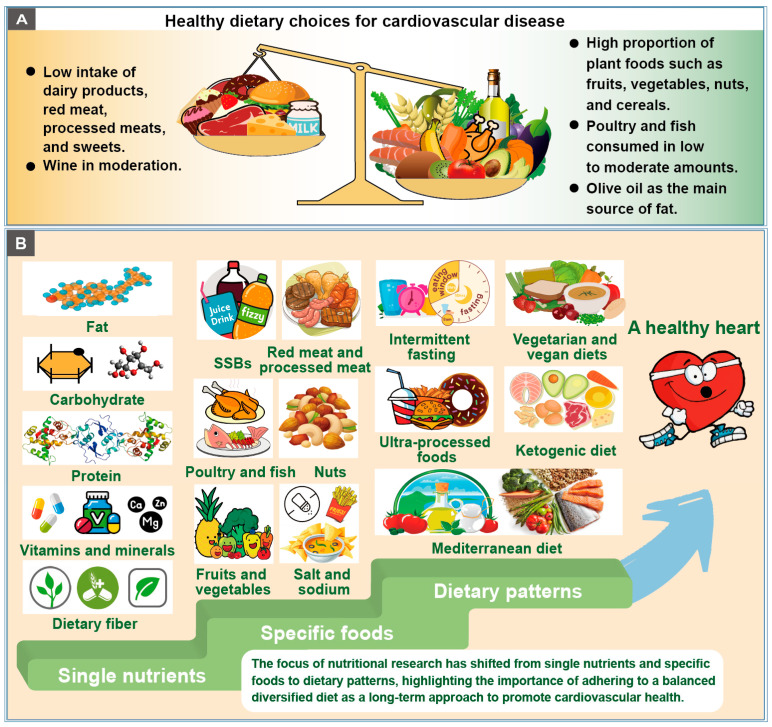
Healthy dietary choices to promote cardiovascular health. (**A**) Dietary patterns; (**B**) focus of nutritional research for CVD has shifted from single nutrients and specific foods to dietary patterns.

**Table 2 nutrients-15-04898-t002:** Summary table of the 2021 ESC and AHA dietary guidance for CVD prevention.

Dietary Component	ESC [218]	AHA [219]
Vegetables and fruits	≥200 g of vegetables per day (≥2–3 servings). ≥200 g of fruit per day (≥2–3 servings).	Eat plenty of fruits and vegetables, and choose a wide variety.
Unsalted nuts	30 g unsalted nuts per day.	
Grains	Carbohydrates from whole grains. 30–45 g of fiber of per day, preferably from whole grains.	Choose foods made mostly with whole grains rather than refined grains.
Fatty	Saturated fatty acids should account for <10% of total energy intake, through replacement by PUFAs, MUFAs. Trans unsaturated fatty acids should be minimized as far as possible, with none from processed foods.	Use liquid plant oils rather than tropical oils (coconut, palm, and palm kernel), animal fats (e.g., butter and lard), or partially hydrogenated fats.
Protein		a. Mostly protein from plants (legumes and nuts). b. Fish and seafood. c. Low-fat or fat-free dairy products instead of full-fat dairy products. d. If meat or poultry are desired, choose lean cuts and avoid processed forms.
Salt	<5 g total salt intake per day.	Choose and prepare foods with little or no salt.
Meat	Red meat should be reduced to a maximum of 350−500 g a week; in particular processed meat should be minimized. Fish is recommended 1–2 times per week, in particular fatty fish.	
Alcoholic beverages	Consumption of alcohol should be limited to a maximum of 100 g per week.	If you do not drink alcohol, do not start; if you choose to drink alcohol, limit intake.
SSBs	Sugar-sweetened beverages, such as soft drinks and fruit juices, must be discouraged.	Minimize intake of beverages and foods with added sugars.
Ultra-processed foods		Choose minimally processed foods instead of ultra-processed foods.
Dietary patterns	Adopt a more plant-based and less animal-based food pattern.	

## Data Availability

Data are contained within the article.

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
