# Peer review of "A Review of Healthy Dietary Choices for Cardiovascular Disease: From Individual Nutrients and Foods to Dietary Patterns"

_nutrients, 2023, doi:10.3390/nu15234898_

Round 1
Reviewer 1 Report
Comments and Suggestions for Authors
In an extensive review, dr. Wenjing Chen and colleagues performed “…an up-to-date summary of knowledge from population-based randomized control trials (RCTs), observational studies, and meta-analyses in an effort to comprehensively examine the role of specific diet components and dietary patterns in influencing CVD risk and health outcomes”.
The review is well-organized, generally well-written (with some typos and errors that should be corrected) and comprehensive.
Actually, I see a couple of points to ameliorate it.
1) As the authors know most of the knowledge about food and CV disease come from large observational studies and meta-analyses, but there are only a few clinical trials, that conceptually are superior, to certify the superiority of a diet/macronutrients/food with respect to another. Still, in this kind of research there are multiple confounding factors and possible bias about data collections….I would suggest that the authors, prior to enter in depth in the argument, put a paragraph where try to critically summarize these points.
2) Some important foods, such as yogurt and potatoes, maybe deserve a more detailed analysis. I would suggest to mention at least the products that are mentioned in this recent paper (see beyond), if not already present in the review.
O'Hearn M, Lara-Castor L, Cudhea F, Miller V, Reedy J, Shi P, Zhang J, Wong JB, Economos CD, Micha R, Mozaffarian D; Global Dietary Database. Incident type 2 diabetes attributable to suboptimal diet in 184 countries. Nat Med. 2023 Apr;29(4):982-995. doi: 10.1038/s41591-023-02278-8. Epub 2023 Apr 17. PMID: 37069363; PMCID: PMC10115653.
Comments on the Quality of English LanguageThere are some typos and errors that should be corrected.
Reviewer 2 Report
Comments and Suggestions for Authors
First, I would like to extend my congratulations to the authors on crafting such a comprehensive review. The choice of topic is exceptionally relevant and timely, particularly in an era where there is burgeoning interest in various fad diets and their purported health benefits, despite the minimal evidence underpinning many of these beliefs.
However, to further improve the readability and usefulness of the article, I have several suggestions:
- Clarification of Terms:
- The term "healthy dietary choice" is used frequently throughout the manuscript but remains undefined. It would be beneficial for the readers if the authors could provide a clear definition or describe what this term encompasses within the context of this review.
- Addressing Limitations:
- While the article discusses healthy diets, it neglects to address the real-world limitations of adopting such diets. The theory of healthy eating is often straightforward, but the practical implementation can be fraught with challenges. Factors such as cultural and geographical influences, economic barriers, access to healthy foods, and the importance of early dietary intervention from a young age are all critical considerations that warrant discussion.
- Diversity of Dietary Definitions:
- The lack of a uniform or universal definition for the nutritional composition of fad diets, such as the Mediterranean or ketogenic diets, is a significant point. This variance can lead to disparate conclusions across different studies. It would be prudent for the authors to acknowledge this issue and its impact on research outcome consistency.
- Guidelines in Clinical Practice:
- Assuming that this article may serve as a foundation for future studies, it would be instrumental to include a discussion on existing guidelines in clinical practice. Which guidelines from various authoritative bodies, such as the nutrition committee, cardiology, or obesity societies, are currently in use, and are there any discrepancies among them?
- Implementation Strategies:
- Understanding that raising awareness, managing cost, and overcoming other challenges are significant hurdles, the authors should delve into potential strategies for implementing these dietary recommendations in real-world settings. Also, add few sentences on how current public health policies support or hinder healthy dietary choices, especially in relation to CVD, could provide a valuable societal context to the discussion.
6. Future Research Directions:
- Finally, suggestions for future research in the field of nutrition and CVD health could be provided, highlighting gaps in the current literature that need to be addressed. The authors might consider suggesting the need of long-term longitudinal studies or dietary intervention trials that provide insight into the long-term effects of certain diets on CVD health.
I look forward to the authors' response to these comments and to a revised manuscript that includes these important enhancements.
Round 2
Reviewer 2 Report
Comments and Suggestions for Authors
I would like to extend my congratulations to the authors for thoroughly addressing the reviewer's comments. I now endorse this manuscript for publication.